# The Effect of Combined Training and Racing High-Speed Exercise History on Musculoskeletal Injuries in Thoroughbred Racehorses: A Systematic Review and Meta-Analysis of the Current Literature

**DOI:** 10.3390/ani10112091

**Published:** 2020-11-11

**Authors:** Kylie L. Crawford, Benjamin J. Ahern, Nigel R. Perkins, Clive J. C. Phillips, Anna Finnane

**Affiliations:** 1School of Veterinary Science, The University of Queensland, Gatton 4343, Australia; b.ahern@uq.edu.au (B.J.A.); n.perkins1@uq.edu.au (N.R.P.); 2School of Public Health, The University of Queensland, Herston 4006, Australia; a.finnane@uq.edu.au; 3Curtin University Sustainability Policy (CUSP) Institute, Curtin University, Perth 6845, Australia; Clive.Phillips@curtin.edu.au

**Keywords:** racehorse, thoroughbred, epidemiology, musculoskeletal, injury, wastage

## Abstract

**Simple Summary:**

Despite extensive international research, musculoskeletal injuries remain a key problem for the Thoroughbred racing industry, with multiple welfare and ethical consequences. High-speed exercise (HSE) history is an important risk factor for injury. However, studies report conflicting findings concerning the effect of HSE on the risk of injury. Although most injuries and fatalities occur during training rather than racing, many research studies only evaluate injuries reported on race day, which likely underrepresents the injuries acquired by these horses. This study aimed to determine the effect of combined training and racing HSE on injuries in racehorses. We performed a systematic search of the relevant literature and evaluated the effect of seven measures of HSE on injury through meta-analyses. The total career HSE distance significantly affected the odds of musculoskeletal injuries (MSI). With every 5-furlong (1 km) increase in career HSE distance, the odds of MSI increased by 2%. The average HSE distance per day also affected the odds of MSI. With every additional furlong (200 m) of average HSE per day, the odds of MSI increased by 73%. However, the strength of the available evidence is restricted by methodological limitations, which will need to be addressed in the future, to explore the level of risk and mechanisms for injury in more detail.

**Abstract:**

Despite over three decades of active research, musculoskeletal injuries (MSI) remain a global problem for the Thoroughbred (TB) racing industry. High-speed exercise history (HSEH) has been identified as an important risk factor for MSI. However, the nature of this relationship remains unclear, with an apparent protective effect of HSE against injury, before it becomes potentially harmful. Many MSI cases and fatalities occur during training rather than during racing, resulting in an underestimation of injury from studies focused on race day. The objective of this study was to examine the current evidence of the effect of combined training and racing HSEH on MSI in TB flat racehorses, through a systematic review and meta-analysis. A systematic search of the relevant literature was performed using PubMed^®^, Scopus^®^, Web of Science^®^, and Embase^®^ online databases and the gray literature using sites containing “.edu” or “.edu.au”. Studies included in the review had explored seven different measures of HSE, including total career HSE distance, cumulative HSE distance in the 30 and 60 days before MSI, average HSE distance per day, per event and per 30 days, and the total number of HSE events. The total cumulative career HSE distance significantly affected the odds of MSI, with every 5-furlong increase, the odds of MSI increased by 2% (OR = 1.02; 95% CI 1.01, 1.03; *p* = 0.004). The average HSE distance per day also affected the odds of MSI, with every additional furlong increasing the odds of MSI by 73% (OR = 1.73; 95% CI 1.29, 2.31; *p* < 0.001). Other measures of HSE were not found to be consistently associated with risk of MSI, but these results should be interpreted with caution. Significant methodological limitations were identified and influence the comparability of studies. Standardizing the measures of HSE in studies of MSI, and describing training conditions in more detail, would support a more thorough investigation of the relationship between HSE and MSI. An improved understanding of this relationship is critical to mitigating the impact of MSI in the Thoroughbred racehorse.

## 1. Introduction

Despite over three decades of active research, musculoskeletal injuries (MSI) remain a global problem for the Thoroughbred (TB) racing industry [1,2,3]. There are important ethical, welfare, and economic consequences resulting from MSI. A principal issue is the serious injury and/or death of horses [4,5,6,7,8] and riders [9,10]. A further important consequence of MSI is the involuntary retirement of these horses from racing, which results in substantial economic losses [7,11] and public concern regarding the final destination/repurposing of these horses [4,12,13,14,15,16].

There has been and continues to be a strong interest in developing training and management strategies to reduce the impact of MSI. High-speed exercise history (HSEH) has been identified as an important risk factor for MSI due to the high loads and strains generated [17,18,19]. However, the association reported between HSEH and MSI is inconsistent. Some studies report that as HSEH increases [20,21,22,23,24,25,26,27,28,29], the risk of MSI increases, while others report it decreases [22,30,31,32,33,34,35,36,37] or does not change [37,38,39,40]. Furthermore, other studies report increasing HSEH has a second order effect on the risk of MSI, whereby as HSEH continues to increase the risk of MSI initially decreases, plateaus, and then increases again [29,31,33,37]. Possible reasons for inconsistent findings include differences in case (outcome) definitions, study populations, geographical locations and associated training conditions, the way HSEH is reported, the complex relationship between HSEH and MSI, or any combination of these factors.

Many MSI cases and fatalities occur during training rather than during racing [23,26,41,42,43,44]. Therefore, studies with a case definition of race day MSI will miss a large proportion of cases that occur during training. These studies will also not capture MSI cases that are not apparent on the day of racing and are discovered later [42,45]. Analyzing training data may be more beneficial than analyzing racing data as it captures a larger exposure to HSEH and a larger number of MSI. This may more accurately represent the impact of HSEH on MSI. Furthermore, modifications to reduce the impact of MSI are more readily implemented at the training level.

A recent meta-analysis of risk factors for catastrophic MSI in flat racing reported that a greater number of race starts increased the risk of catastrophic MSI, but the effect of cumulative distances HSEH were inconsistent [46]. There is a need to quantify the relationship between HSEH and MSI, focusing on MSI incurred during training and all MSI, not only catastrophic injuries. By quantifying the relationship between HSEH and MSI training programs can be modified to reduce the incidence of MSI. Thoroughbred racing is a highly contentious, high profile, and lucrative industry and as such attracts significant media and public opinion. It is vital that independent research is conducted to understand and mitigate the risks of MSI. We aimed to determine the effect of combined training and racing HSEH on MSI in TB flat racehorses.

## 2. Materials and Methods 

### 2.1. Research Question

We examined the research question “What is the effect of the combined training and racing HSEH on MSI in flat TB racehorses?” 

### 2.2. Review Protocol

The protocol for this systematic review and meta-analysis was developed according to the guidelines from the Cochrane Handbook [47], the Preferred Reporting Items for Systematic Reviews and Meta-Analyses (PRISMA) statement [48], and Meta-Analysis of Observational Studies in Epidemiology (MOOSE) reporting guideline [49].

Studies were eligible to be included if they were cohort, case-control, or cross-sectional studies of HSEH and MSI in TB flat racehorses. HSEH was defined as exercise at speeds greater than or equal to 15 s/furlong (13.3 m/s, 800 m/min or 48 km/h) which incorporates “three-quarter pace”, “evens”, “working gallop”, “gallop off the bit”, and “racing speed” [23,29,50]. Seven different measures of HSEH were defined: total career HSE distance, cumulative HSE distance in the 30 and 60 days before MSI, average HSE distance per day, per event and per 30 days, and the total number of HSE events. MSI was defined as any traumatic injury to the musculoskeletal system, incorporating both orthopedic and soft tissue injuries. No limits were placed on date of publication, as no changes to training and management practices have occurred since the first studies on MSI in 1982. 

Articles were identified through searching PubMed^®^, Scopus^®^, Web of Science^®^, and Embase^®^ online databases using the following search terms: Horse*, Equine, Thoroughbred, Racing, Racehorse*, Training, Epidemiolog*, Musculoskeletal injury, Wastage, Fracture*, and Fatal*. The following Medical subject heading (MeSH) terms were also incorporated into the search strategy: Horses, Equidae, Epidemiology, and Cumulative Trauma Disorders. The search strategy is further detailed in Appendix A. Literature searches were performed in April 2019. Articles were screened for relevance independently by two investigators (KC and BA), and any discrepancies in study selection were resolved by consensus. Articles published in peer-reviewed journals, conference proceedings, and dissertations were considered for inclusion. 

Data were extracted by KC and BA, including authors, title, year of publication, country, study type, population source, outcome definition, selection of individuals without the outcome, number of individuals with the outcome, number of individuals without the outcome, the HR/RR/OR from univariable regression analysis, 95% confidence intervals, and the *p*-values. Where sufficient data could not be extracted from the manuscript, for example, where an effect size was reported but not the total numbers of events, authors were contacted for clarification. When more than one outcome category was available, the category which provided the largest number of individuals with the outcome was extracted for analysis. When race, timed workout, and combined race and workout data were provided, the combined data were extracted for analysis. When active career and total career data were provided the active career data were extracted for analysis. 

The incidence rate ratio (relative risk) (IRR/RR) and odds ratios (OR) were considered equivalent effect sizes (ES) for this analysis by virtue of the rare outcome assumption. The OR approximates the RR when the probability of the event is less than 0.10, which holds true for horses in all included studies and for all exposures to HSE [51]. When the ES was provided per meter, this was converted to per furlongs by multiplying the natural log of the effect size by 200. Data that had been categorized due to non-linearity [29,31,44], was excluded from the meta-analysis because the effect size between categories was not comparable to the effect size between a 1-furlong (200 m) increment increase in HSEH. A minimum of two studies with a comparable effect size were considered sufficient for meta-analyses to be performed [52].

Risk of bias assessment was performed by two investigators (KC and BA), using relevant items from the Cochrane tool for assessing risk of bias [47] and the Newcastle Ottawa Scale [53]. Methods for selecting trainers and horses, definition and measurement of exposure and outcome variables, data completeness and follow-up, selective reporting, and declaration of conflicts of interest were all considered in terms of their potential for introducing bias.

Meta-analyses were performed by KC using Stata 15.1^®^ (Statacorp). Separate analyses were performed for each measure of HSEH exposure, using a random effects model described by DerSimonian and Laird [54]. The random effects model was necessary to account for other sources of variability between studies than HSEH. Cochrane’s Q test statistic was used to test for heterogeneity [55]. A cut-off of *p* < 0.10 was used to ameliorate the possibility of a type 1 error due to low numbers of studies [55]. Higgins I^2^ was used to evaluate the proportion of between study variation in ES that was due to true heterogeneity rather than sampling variability. Confidence intervals for I^2^ were calculated when there were greater than 2 degrees of freedom for Cochrane’s Q test statistic. We interpreted the I^2^ values of 0% to 30% to be not clinically important, 31–60% to represent moderate heterogeneity, 60–75% to represent substantial heterogeneity, and >75% to represent high heterogeneity [56]. 

## 3. Results

### 3.1. Screening of Articles

The articles identified, screened, and assessed as relevant and eligible for inclusion in the review are summarized in a PRISMA flow diagram 49 (Figure 1). The characteristics of the 181 full text articles screened and the reasons for exclusions are presented in Appendix A. 

A graphical summary of the articles excluded at full text level and the reasons for exclusion are presented in Figure 2. 

Exclusion Criteria: Review articles/editorials, case series/case reports, clinical trials, descriptive not analytical studies, or abstracts only (n = 86);Reported only race day rather than training data (n = 17);The outcome (MSI) was reported as morphological changes to bone/soft tissue, rather than as a clinical MSI (n = 49);The exposure, high-speed exercise (HSE) was not reported as total career cumulative HSE distance, cumulative HSE distance 30 days before MSI, cumulative HSE distance 60 days before MSI, average HSE distance per day, average HSE distance per event, average HSE distance per 30 days, or total number of HSE events;The effect size was not reported as either an odds ratio, relative risk, or hazard ratio (n = 143).

### 3.2. Study Characteristics

Thirteen studies examined the effect of HSEH on MSI, including one cross-sectional, seven case-control (including two nested case-control), and five prospective cohort studies. Study characteristics for all eligible studies are detailed in Appendix A. For any single measure of HSEH, there were between two and five studies assessing HSE exposure using that method and reporting results which could be pooled in meta-analyses. Of the 13 eligible studies in the review, three studies [29,31,44] reported HSEH categorized according to level of exposure and subsequently could not be included in the pooled analyses for that exposure. 

### 3.3. Risk of Bias Assessment

The risk of bias (RoB) assessment indicated that overall, the 13 studies meta-analyses had low risk of bias, with nine criteria assessed as a low risk in all studies, four criteria with at least half the studies assessed as low risk, and only three criteria where a significant number, or all studies, were assessed as having unclear or high risk of bias, including demonstrating MSI was not present at start of study, outcome assessor being aware of the level of exposure, and exercise data completeness. Results from the RoB assessment are presented in Figure 3.

### 3.4. Meta-Analyses

The seven different measures of HSE exposure were grouped into three categories as follows: (1)Cumulative HSE distance:Total cumulative HSE distance;HSE distance accumulated 30 days before injury;HSE distance accumulated 60 days before injury.(2)Rate of HSE distance accumulation:Average HSE distance per day;Average HSE distance per event;Average HSE distance per 30 days.(3)The total number of accumulated HSE events.

Within each of these categories, some studies performed time to event analyses and reported effect sizes as hazard ratios, while others reported odds ratios. These were pooled independently, resulting in a total of ten separate meta-analyses to explore all types of HSE exposure. 

#### 3.4.1. The Effect of Cumulative HSE Distance on Risk of MSI 

Four meta-analyses were performed to investigate the measures of cumulative HSE distance on MSI (Figure 4).

There were five studies examining the effect of the total career cumulative HSE distance on MSI and reporting odds ratios as the effect size [21,57,58,59,60]. Studies were conducted in the USA (number of studies = 3), Japan (n = 1), and Australia (n = 1). The outcomes measured were fatal proximal sesamoid bone fracture/catastrophic suspensory apparatus failure (n = 2), catastrophic MSI (n = 1), superficial digital flexor tendonitis (n = 1), and stress or complete fractures of the humerus (n = 1). The total cumulative HSE distance significantly affected the odds of MSI (OR 1.02; 95% CI 1.01 to 1.03; *p* = 0.004) (Figure 4). With every increase of 5 furlongs (1 km) of total cumulative HSE, the odds of MSI increased by 2%. The tau-squared (absolute value of the true variance (heterogeneity) of the effect size) was 0.0000; Cochrane’s Q was 5.72 (*p* = 0.22); and I^2^ (the proportion of the variation in effect size attributed to true heterogeneity rather than sampling error) was 30% (95% CI 0% to 73%). This indicated that the heterogeneity was low and not significant.

There were three studies examining the effect of the total career cumulative HSE distance on MSI, using time to event analyses and reporting HR as the effect size [28,40,61]. These studies were conducted in New Zealand (n = 1) and the UK (n = 2). The outcomes measured were MSI (n = 1), carpal joint disease (n = 1), and dorsal metacarpal disease (n = 1). With every increase of 5 furlongs (1 km) of total cumulative HSE, the HR of MSI increased by 9%, although this was not significant (HR 1.09; 95% CI 0.99, 1.19; *p* = 0.08) (Figure 5). The tau-squared was 0.003; Cochrane’s Q was 2.77 (*p* = 0.25); and I^2^ was 28% (95% CI 0%, 92%). This indicated the heterogeneity was low and not significant. 

Five studies, from the US (n = 4) and NZ (n = 1), examined the effect of the HSE distance accumulated in the 30 days prior to injury on MSI, and reported odds ratios [21,31,33,57,58]. Injury types were fatal proximal sesamoid bone fracture/catastrophic suspensory apparatus (n = 2), catastrophic MSI (n = 1), lower limb MSI (n = 1), and catastrophic fractures of the humerus (n = 1). With every 1 furlong increase in the HSE distance accumulated in the 30 days before the MSI occurred, the pooled odds of MSI decreased by 3%, although this was not significant (OR 0.97; 95% CI 0.90 to 1.02; *p* = 0.36) (Figure 6). The tau-squared was 0.01, Cochrane’s Q was 95.99 (*p* < 0.001), and I^2^ was 96%, (95% CI 93% to 98%) indicating high and significant heterogeneity. 

The four studies from the USA which examined HSE in the 30 days prior to MSI also explored HSE distance in the 60 days prior to MSI, reporting the effect size as odds ratios [21,31,57,58]. The outcomes measured were fatal proximal sesamoid bone fracture/catastrophic suspensory apparatus (n = 2), catastrophic MSI (n = 1), and catastrophic fractures of the humerus (n = 1). With every 1 furlong increase in the cumulative HSE distance in the 60 days before the MSI occurred, the odds of MSI decreased by 1%, although this was not significant (OR 0.99; 95% CI 0.99 to 1.04; *p* = 0.65). The tau-squared was 0.002, Cochrane’s Q was 45.22 (*p* < 0.001), and I^2^ was 93%, (95% CI 86%,97%) indicating high and significant heterogeneity.

#### 3.4.2. The Effect of the Rate of HSE Distance Accumulation on Risk of MSI 

Five meta-analyses were performed to investigate the measures of the rate of HSE distance accumulation on MSI (Figure 5).

There were two studies exploring the effect of average HSE distance per day on MSI, using odds ratios as the effect size [31,57]. The outcomes measured were catastrophic suspensory apparatus failure (n = 1) and catastrophic scapular fractures (n = 1). The average HSE distance per day significantly affected the odds of MSI (OR 1.73; 95% CI 1.29 to 2.31; *p* < 0.001). With every increase of 1 furlong of HSE per day, the odds of MSI increased by 73%. The tau-squared was 0.0000, Cochrane’s Q was 0.24 (*p* = 0.62), and I^2^ was 0%, indicating the heterogeneity was low and not significant.

A further two studies from the UK also examined the effect of the average HSE distance per day on MSI using hazard ratios as the effect size [28,61]. Injury types were metacarpophalangeal/metatarsophalangeal joint disease (n = 1) and dorsal metacarpal disease (n = 1). With every increase of 1 furlong in the average HSE distance per day, the HR of MSI increased by 19%, although this was not significant (HR 1.19; 95% CI 0.91 to 1.57; *p* = 0.20). The tau-squared was 0.03, Cochrane’s Q was 7.83 (*p* = 0.01), and I^2^ was 87%, indicating the heterogeneity was substantial and significant.

Five studies examined the effect of the average HSE distance per event on MSI using odds ratios as the effect size [21,31,58,59,62]. Contributing studies were from the USA (n = 3), Australia (n = 1), and Japan (n = 1) and the injuries evaluated were fatal proximal sesamoid bone fracture (n = 1), MSI (n = 1), catastrophic MSI (n = 1), superficial digital flexor tendonitis (n = 1), and catastrophic scapular fracture (n = 1). With every increase of 1 furlong in the average HSE distance per event, the odds of MSI increased by 16%, although this was not significant (OR 1.16 (95% CI 0.92 to 1.45; *p* = 0.21)). The tau-squared was 0.05, Cochrane’s Q was 91.01 (*p* < 0.001), and I^2^ was 96% (95% CI 92%, 98%), indicating high and significant heterogeneity. 

Two of the studies examining HSE distance per event also evaluated the effect of the average HSE distance per 30 days on MSI using odds ratios as the effect size [21,58]. These studies were both from the USA and they reported on fatal proximal sesamoid bone fractures (n = 1) and catastrophic MSI (n = 1). With every increase of 1 furlong in the average HSE distance per 30 days, the odds of MSI decreased by 3%, although this was not significant (OR 0.97 (95% CI 0.85 to 1.10; *p* = 0.60)). The tau-squared was 0.01, Cochrane’s Q was 6.97 (*p* = 0.01), and I^2^ was 86%, indicating high and significant heterogeneity. 

A further two studies from the UK explored the effect of the average HSE distance per 30 days on MSI using time to event data and reporting hazard ratios as the effect size [28,61]. The injury types evaluated were metacarpophalangeal/metatarsophalangeal joint disease (n = 1) and dorsal metacarpal disease (n = 1). With every increase of 1 furlong in the average HSE distance per 30 days, the HR of MSI increased by 7%, although this was not significant (HR 1.07 (95% CI 0.99 to 1.16; *p* = 0.11)). The tau-squared was 0.003, Cochrane’s Q was 6.28 (*p* = 0.01) and I^2^ was 84%, indicating high and significant heterogeneity.

## 4. Discussion

We quantified the effect of HSEH on MSI using seven measures of HSE exposure that were broadly classified into cumulative HSE distance, the rate of HSE distance accumulation, and the number of accumulated HSE events. Of the three measures of cumulative HSE distance, the total career cumulative HSE distance significantly increased the odds of MSI. Of the three measures of the rate of HSE distance accumulation, the average HSE distance accumulated per day significantly increased the odds of MSI. The heterogeneity (variance in effect sizes between studies) observed was low, supporting the reliability of these findings.

The remaining measures of cumulative HSE distance, the rate of HSE distance accumulation, and the number of accumulated HSE events were not found to significantly affect the risk of MSI. However, caution should be exercised when interpreting these findings. The significant heterogeneity indicates that we are not comparing equivalent studies, and the variability between studies is too high [63]. This high variability is preventing a reliable assessment of the effect of these measures of HSEH on MSI. We attempted to identify potential sources of heterogeneity but there was no discernible pattern between these studies. Formal subgroup analyses and meta-regression could not be performed due to insufficient numbers of studies in the relevant subgroups [47,52,63]. Significant heterogeneity was also reported in a previous meta-analysis of catastrophic racing MSI [46].

Possible explanations for the heterogeneity in this study included differences in study design, populations of horses, geographic location, outcome definitions, and methods to determine the outcomes. Study designs in this meta-analysis were cross-sectional [57], case-control [21,31,58,59,60], nested case-control [29,44], and prospective cohort [28,33,40,61,62]. Populations of horses were two-year-old horses [28,40,62] and horses of all age groups [29,31,33,57,59]. Geographic locations were the UK [28,29,44,61], USA [21,31,57,58], Japan [59], New Zealand [33,40], and Australia [60,62]. Outcome definitions were fatal proximal sesamoid bone fracture [21], catastrophic suspensory apparatus failure [57], catastrophic scapula fracture [31], catastrophic MSI [58], lower limb MSI [33], all types of MSI [40] fracture [29], stress or complete humeral fractures [60], fetlock joint injuries [61], superficial digital flexor tendon injuries [59], and dorsal metacarpal disease [28]. Methods used to determine the outcome also varied between studies. These methods were analyzing racing databases, results of post-mortem studies, trainer reported injuries, and veterinary examinations. However, we could not identify any pattern to these differences in study designs, geographic locations, or outcome definitions that could explain these observed differences in results.

Geographic location is likely a major source of variability between studies, despite no consistent pattern was observed in these meta-analyses. Differing management practices between racing jurisdictions influence the accessibility and nature of available data. For example, racehorses in Japan and Hong Kong are accommodated, trained, and raced at centers tightly controlled by racing authorities [64,65]. Consequently, all HSE data is recorded. Racing authorities in the US record timed works, which are almost exclusively held at the racetrack [19,24,31,66]. Similarly, Australian racing authorities record official race trials, which are frequently held on training tracks rather than on racetracks [26,67,68]. However, a large proportion of HSE is performed during unofficial trials or training, which is not recorded [26,67,68]. Similarly, a large proportion of HSE in the UK is performed at training centers or private training yards and is not recorded [43,44,69]. Furthermore, training practices may also vary with geographic location, both between countries and regions within a country. These discrepancies in training and racing conditions and the subsequent effect on data completeness are likely to further complicate efforts to isolate the effect of HSE on the risk of MSI.

The disparity in findings between studies may also be due to a second order effect of HSE on MSI, whereby the risk of MSI could move in any direction depending on the level of HSE. We hypothesize that it initially decreases with increasing HSE, plateaus, and then increases again after a critical level of HSE has been reached. Such a relationship has been reported in studies of racehorses [29,31,33,37] and human athletes [70,71]. This relationship between HSEH and MSI is likely due to two mechanisms of injury: (1) low training intensity in poorly adapted tissue; and (2) high training intensity in well adapted tissue [46]. HSE is required for the tissue adaptation necessary to prevent MSI [30]; however, beyond a critical point, HSE is an important risk factor for tissues failing [70,71,72,73]. If there is a second order relationship between HSE and MSI, the overall observed effect size would depend on where individual horses are on this curve. For example, if one horse was at high risk of MSI due to a low exposure to HSE and another horse was at a low risk of MSI because it had sufficient HSE for tissue adaptation, yet not sufficient to cause degenerative changes, their combined overall effect size would be towards the null. Thus, the magnitude of any overall linear association between HSE and MSI could be diluted by this second order relationship. We were unable to investigate this possibility in the current meta-analysis because no studies provide the required data. 

Our findings are consistent with this second order relationship. The positive linear association between increasing total cumulative HSE distance and MSI in analysis 1 is consistent with excessive HSE in well-adapted bone. This is plausible as most horses in these studies were older horses and were likely to have had sufficient HSE for bone adaptation, and further increases in HSE resulted in increased odds of MSI. Similarly, in analysis 5, the positive association between increasing average HSE per day and increasing odds of MSI could also be consistent with increasing HSE overloading the already adapted tissues. 

The non-significant association between the other five measures of HSEH and MSI in the current meta-analysis may also reflect a second order relationship and the two above mentioned mechanisms of injury. For example, approximately half the studies reported increasing the cumulative distances of HSE 30 and 60 days before MSI decreased the risk of MSI, and half of the studies reported it increased the risk. This could hold true in both general ends of the second order association between HSE and injury. In early training, with poorly adapted tissues, elevated HSE distance on a given day may overwhelm tissue tolerance and lead to injury. In later training with adapted tissues, excessive average HSE may still exceed tissue tolerance but the threshold for injury may be either higher, or potentially impacted by lifetime cumulative HSE, leading to microdamage and an increased susceptibility to injury. These polarized findings could reflect these two mechanisms of injury. Investigating the length of time before HSE is undertaken when commencing training or resuming from a break in training, reduction in training and racing intensity or a sudden disruption in the full training program could also help define the amount of HSE required for tissue adaptation and the point beyond which it is harmful. 

The predominant limitation of this study is the heterogeneity observed between studies preventing a meaningful interpretation of the effect of several measures of HSE on MSI. Our stringent exclusion criteria have also excluded studies containing generic data with surrogate measures of HSE exposure such as age or length of time in training. Exploring the effect of HSE in horses of different ages was also not possible as this information was not available.

This study highlighted the need for a detailed investigation into the epidemiology of training practices and how they vary between geographic regions. There is a need to capture this information in a standardized manner, analogous to the European consensus on reporting the epidemiology of injuries of jockeys and racing industry participants [74]. This would facilitate comparison between studies and enable large multi-centered studies to be conducted. Improving both comparability of studies and study power would better elucidate the relationship between HSEH and MSI [75]. Determining the level of HSE required for tissue adaptation and the critical point beyond which further HSE is harmful could considerably reduce the impact of MSI. 

We recommend that future studies use a prospective study design to obtain the best possible quality data, although they require more time and resources to conduct [76,77,78]. Retrospective studies inherently contain errors due to incomplete or misclassified data [79]. Cohort studies have the additional advantage of incorporating time at risk into the analysis [76]. Both training and race day data should be collected to capture the total HSEH exposure and MSI outcomes [23,26,41,42,43,44]. Detailed information regarding the distance covered at different velocities by horses during training and racing and regarding the surface characteristics of training and racing tracks must be captured, as this information is often not available [75]. There is a lot of variety and secrecy around training methods, and preserving trainer confidentiality is paramount to collecting accurate, detailed training data. Finally, data must be coordinated and presented in a manner that trainers can relate to, as ultimately, they have the greatest potential to reduce MSI.

## 5. Conclusions

In conclusion, increasing HSE increases the odds of MSI. The measures of HSE distance that best capture this relationship are the total career cumulative HSE distance and the average HSE distance accumulated per day. The remaining measures of HSE distance, rate of accumulation, and the number of events could not be properly evaluated due to the heterogeneity observed in the meta-analyses. Consequently, we recommend that a detailed investigation into the epidemiology of training practices and how they vary between geographic regions is required. This information must be standardized to facilitate comparability between studies and multi-center collaboration, which would better elucidate the relationship between HSE and MSI. An improved understanding of this relationship is critical to mitigating the impact of MSI in the Thoroughbred racehorse. 

## Figures and Tables

**Figure 1 animals-10-02091-f001:**
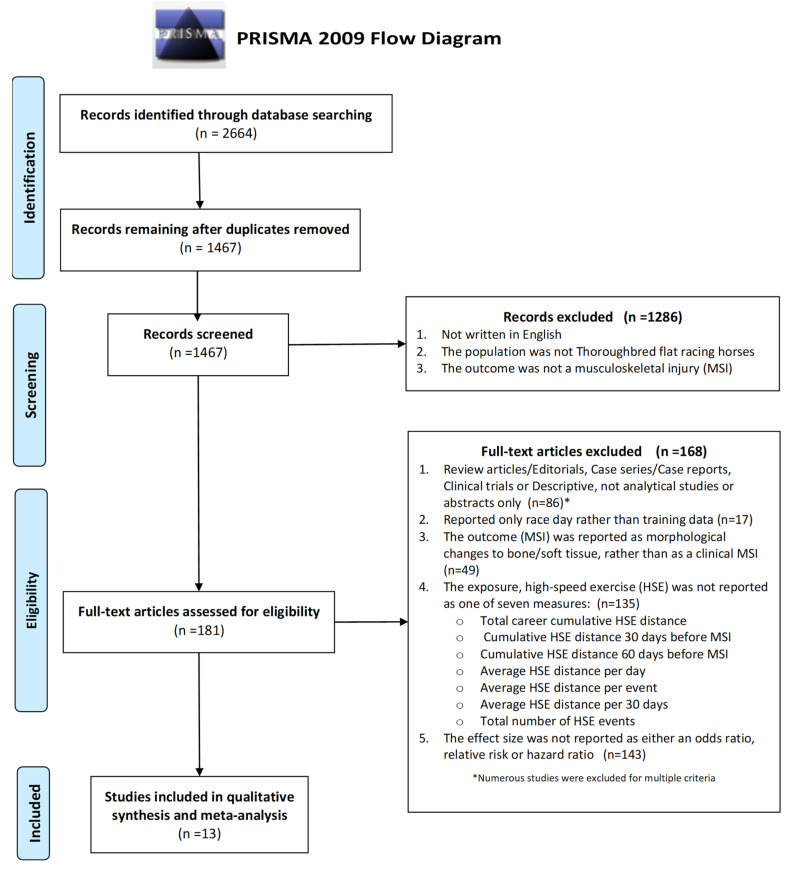
The identification, screening, eligibility, and included articles summarized in a Preferred Reporting Items for Systematic Reviews and Meta-Analyses (PRISMA) flow diagram.

**Figure 2 animals-10-02091-f002:**
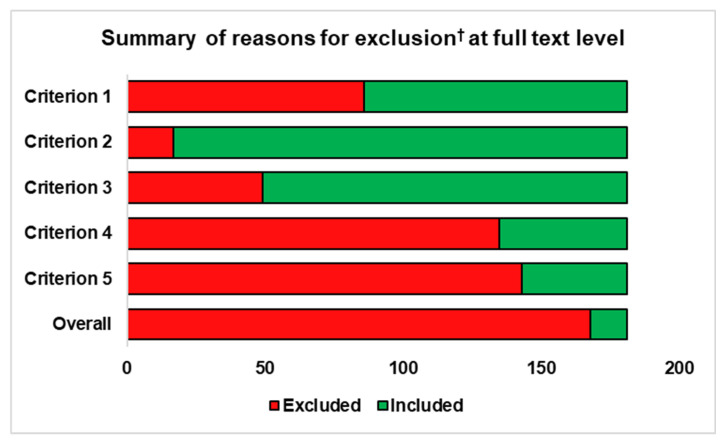
Summary of the secondary screening of full text articles investigating the effect of high-speed exercise history (HSEH) on musculoskeletal injury (MSI).

**Figure 3 animals-10-02091-f003:**
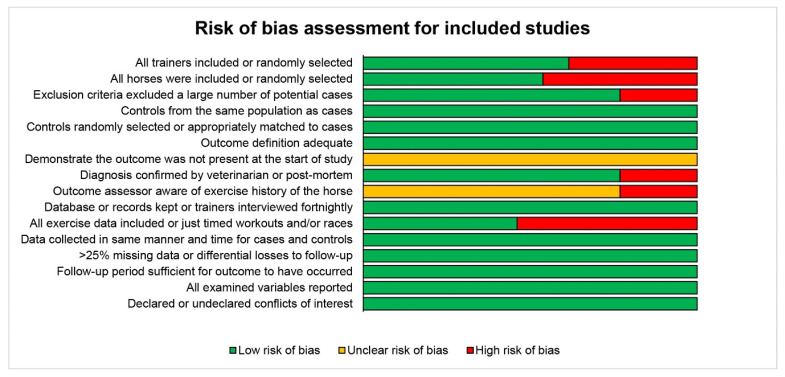
The risk of bias assessment for included studies.

**Figure 4 animals-10-02091-f004:**
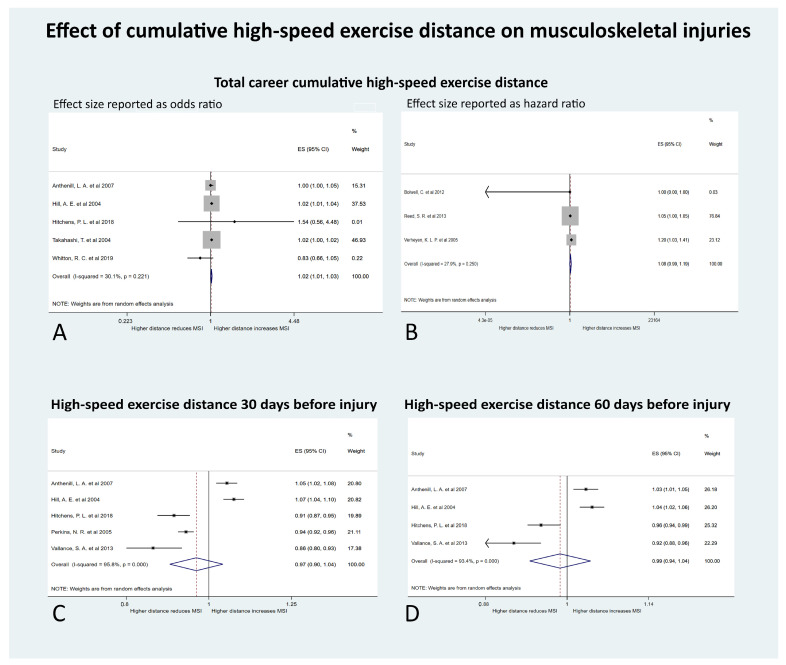
Forest plots of the effect of measures of cumulative high-speed exercise (HSE) distance on musculoskeletal injuries (MSI) in Thoroughbred racehorses (**A**). Total career cumulative HSE distance with effect size (ES) reported as odds per increase of 5 furlongs (1 km) of HSE history (**B**). Total career cumulative HSE distance with ES reported as hazard per increase of 5 furlongs (1 km) of HSE history (**C**). HSE distance in the 30 days before MSI occurred with ES reported as odds per increase of 1 furlong (200 m) of HSE history (**D**). HSE distance in the 60 days before MSI occurred with ES reported as odds per increase of 1 furlong (200 m) of HSE history.

**Figure 5 animals-10-02091-f005:**
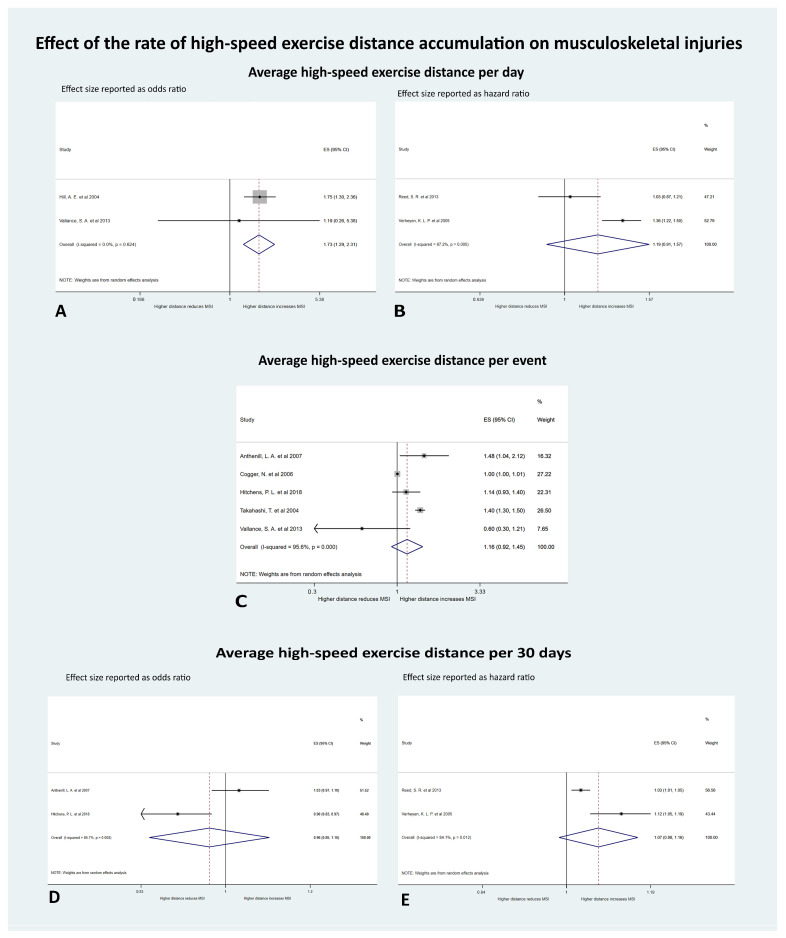
Forest plots of the effect of measures of the rate of high-speed exercise (HSE) distance accumulation on musculoskeletal injuries (MSI) in Thoroughbred racehorses (**A**). Average HSE distance per day with effect size (ES) reported as odds per increase of 1 furlong (200 m) of HSE history (**B**). Average HSE distance per day with ES reported as hazard per increase of 1 furlong of HSE history (**C**). Average HSE distance per event with ES reported as odds per increase of 1 furlong of HSE history (**D**) Average HSE distance per 30 days with ES reported as odds per increase of 1 furlong of HSE history (**E**). Average HSE distance per 30 days with ES reported as hazard per increase of 1 furlong of HSE history.

**Figure 6 animals-10-02091-f006:**
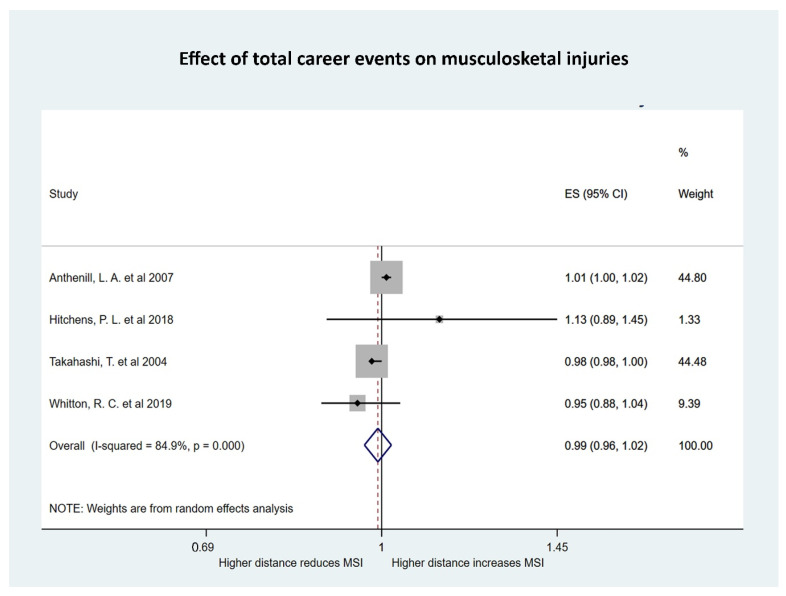
Forest plot of the effect of the total number of accumulated high-speed exercise (HSE) events on musculoskeletal injuries (MSI) in Thoroughbred racehorses with effect size (ES) reported as odds per increase of one high-speed exercise event.

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
