# Peer review of "The Effect of Combined Training and Racing High-Speed Exercise History on Musculoskeletal Injuries in Thoroughbred Racehorses: A Systematic Review and Meta-Analysis of the Current Literature"

_animals, 2020, doi:10.3390/ani10112091_

Round 1

Reviewer 1 Report

A call or uniformity in studies investigating the relationship between exercise and musculoskeletal injury in racehorses: a meta-analysis of the current literature

This is an important undertaking and a very unique approach to motivating the analysis required.  The authors are to be congratulated for their efforts and the huge undertaking trying to structure these ideas.  The paper needs significant work but because of the importance of the question and the challenges associated with the analysis this reviewer believes strongly that the effort is merited.

The paper in the current form is unreadable to most people because of the lack of an overarching framework, which is really the point of the paper!  Part of the challenge in reading this paper is the lack of clarity regarding the different conditions for the meta-analysis.  For example, Figure 4 and Figure 5 have the same figure title and the same first sentence of the caption.  The ES is “odds” versus  “Hazard ratio”.  Instead of just cutting and pasting and leaving it to the reader to slog through this perhaps a topic paragraph and a clarification of the two approaches would make this more readable.  Is that possible within the framework, since both odds and hazard ratio are reasonable ways to present the data.  Figure 8 and 9 and Figures 11 and 12 have the same issue and can highlight the same goal of your paper.  Figure 6 and 7 and more interesting since they obviously should be considered together since the 60 day window includes the 30 day window.  I do not know how to parse these ideas more clearly but I know that having identical titles on figures and nearly identical captions makes the paper hard to read and obscures the points you are trying to make.  While I know that this is a results section and not a discussion, the results organization set up the organizational format for the discussion.  Grouping data in a way that it foreshadows the points you are going to make is a fair way to approach the results section.  A series of 10 figures with 7 unique titles is really too many figures and not enough organization for a readable paper.   

The paper also fails to clearly communicate why an overarching framework does not exist in this portion of the literature.  The sources of this data are not consistent between racing jurisdictions and the way in which the data is gathered is complicated by the different manner in which racing is managed in different countries.  In the US and Japan training primarily occurs under the auspices of the racing authorities at racetracks and training centers overseen by the same regulators that oversee the racetrack.  Timed works are almost exclusively held on the racing surfaces at the racetrack.  In contrast racetracks in New Zealand, Australia and in the United Kingdom are in most cases used  exclusively for racing and training is performed at training centers or private training yards.  The differences between the sources of data at Newmarket which may be the most controlled training environment in the UK and Saratoga which may be the most diverse training environment in North America is profound.  While this may be a part of the discussion it may also be part of the methods.  It seems that a table showing the sources of the data and the selection of the population should be noted.  Different results would also be expected based on geography.  I think that the disparate outcomes may be a result of the differences in training in different locations.  For example, in California the horses tend to do a lot more high speed work between races when compared to other jurisdiction where the horses may do very few timed works between races.  The epidemiology needs to capture this in an accessible form so that the work can be used to communicate to a wider audience.

I appreciate that this work shows that conclusions that can be made are limited, but in some ways the authors are guilty of burying the lede.    The biggest messages may be in lines 401-415 which are well written and insightful.  Lines 417-432 fit into a nice story for the discussion which can be used to plot a path forward.  It may be that with the 10 figrues and 7 titles you are trying to do too much with one paper.  It may be that a separate short paper is needed along the lines of “European consensus on epidemiological studies of injuries in the thoroughbred horse racing industry” by M Turner, C W Fuller, D Egan, B Le Masson, A McGoldrick, A Spence, P Wind, P-M Gadot in Br J Sports Med 2012;46:704–708.  However, such an article would need to address the details of horse management in the different racing jurisdictions which would mean that care would need to be taken to include folks who understood the nuances of both operations and existing training data.  Whereas exceptional and complete training data is available from JRA, the more limited data from Equibase in North America would need to be assessed in the context of differences in training operations where in some places (western Louisiana) training centers may not have be included in timed works.    

As currently organized I cannot recommend publication of this paper.  However, with some organization and expansion of the explanation this paper represents two potentially very important papers to understand the effect of HSE on MSI in the racing Thoroughbred. 

Reviewer 2 Report

ANIMALS #965278

This study applies systematic review and meta-analysis to examine the relationship between measures of work exposure and musculoskeletal injury (MSI) in Thoroughbred horses working on the flat. The authors have done a great deal of work and have been very thorough in their approach to the analysis, yet there are a number of details that need to be addressed and there seems to be a disconnect between the technical rigour of the analysis and the interpretation of the results, as laid out below. There are also a number of technical issues that require attention. This is basically a good piece of work, and I encourage the authors to make changes so that we can all benefit from the findings.

As indicated in the title of the paper and the conclusions, the authors’ primary focus appears to be on the need for uniformity in study design and execution, rather than upon the research question they present at the end of the introduction and at the top of Materials & Methods. They draw valid conclusions at the end of the paper, but do not offer a conclusion on the relationship between high-speed exercise and MSI, and thus do not address the primary research objective. This is a fundamental inconsistency that could nonetheless be addressed fairly easily. I recommend they change the title of the paper to be consistent with the primary research objective, address this objective more fully in the discussion, and present the issue of uniformity as a central conclusion. This would also require adjustment of the abstract and summary. Alternatively, they could divide their results into two papers.

Some comment should be offered on the impact of emphasising central tendencies in each analysis rather than looking at contrasting findings (an exposure both increasing and decreasing injury, depending on the study). This is a potential weakness of meta-analysis in some applications, especially when applied to a small number of studies. This is also a study limitation. This issue is particularly sharply indicated in figure 4, where two studies show major differences in ES, yet the confidence interval for the ES for both includes 1. A further limitation is heterogeneity in the specific MSI types addressed in each paper. The authors make reference to this in the discussion, but not as a study limitation.

The authors should address the potential limitations of their exclusion criteria. In particular, item 4 excludes studies containing generic data in which the exposure is overall time in competition, usually using age as a surrogate exposure measure. Such data may or may not be amenable to meta-analysis, yet it may provide an overarching indication of the relationship between athletic activity and MSI, of relevance to the study’s stated primary objective. A significant component of the literature uses this measure, and would at the least be worthy of mention in the revised discussion with some comparative interpretation. Weaknesses in these age data may be no greater than those associated with issues identified by the authors of varying injury definition and type, study design and exposure definition. Equally, the exposure measures presented in the literature are essentially arbitrary measures of convenience, and are based on the assumption that there is some direct relationship between work and cumulative injury, probable but imperfectly characterised.

The authors elected to confine their study to Thoroughbred horses working on the flat, yet the broad conclusions they draw or that could be drawn from their findings apply to, and in some cases have been explored in, other equine athletes. If the authors were to address the tension between their conclusions and their stated research objective by placing more emphasis on the question of the work-injury relationship, they would find a useful source of supporting and contrasting information that would enrich the discussion, their interpretations, and the value of their findings to readers.

The authors refer to the Cochrane Handbook and the PRISMA statement. These refer to the execution of Systematic Reviews, an essential first step when contemplating meta-analysis. In the interests of accuracy and clarity for those readers still unfamiliar with these approaches, it would be useful if the authors differentiated between the sequencing of these two components to clarify their roles and relationship.

Frequent use is made of the word "adaptation". This is not a strictly correct usage. Adaptation refers to the evolutionary changes that progressively adapt a species to its habitat. More correct terminology would be "tissue adaptation", since the authors are referring to changes within an individual in response to pressures within its local environment. These changes recruit an individual's capacity for active response, which is in turn a reflection of evolutionary adaptation.

Frequent use is made of the word "included" e.g., line 99 - in a systematic review this is not appropriate unless fully supported by addendum material detailing the systematic review process, it’s a little like using the expression "et cetera". As per the PRISMA statement, the criteria by which the search was performed requires more explicit disclosure. It is quite sufficient that these details be provided in addenda to the main paper, but they should be provided. I refer the authors to the PRISMA statements.

There are a number of industry terms ("spell", “evens”, for example), that are used without definition. Bearing in mind the wide possible readership, these terms should either be defined or alternatively avoided and descriptive wording used instead.

There are citations included in the paper, usually in figures, that are not included in the reference list accompanying the text - they should be included. Readers may wish to consult all and any citation included in the text. The fact that the citation appears in the addendum is not sufficient.

In the discussion the authors take up a lot of space repeating results, which is not necessary. That space would be better used to discuss how well their study results address the primary objective. Much of the material in the paragraph starting at line 401 is unnecessary and repetitive.

On line 434 reference is made to a curvilinear relationship. I think the authors mean a "second order effect", which would be the preferred term, perhaps with the definition at the first use. Reference should also be made to the fact that such an effect can move in either direction. Also, numerous studies that performed multivariable analysis have identified such effects, which are not putative but fact.

Line 106 - this sentence requires a little explanation, particularly the notion of "enforced changes to training…".

Line 109 - place the search details in an addendum. Please explain why the word "mortality" was not part of your search. Are you certain that the search engines used did apply "fatality" and "mortality"?

Line 141 - please justify contacting authors. This is neither a component of systematic analysis nor of meta-analysis and represents a potential source of bias.

Line 147 - again, a potential source of bias. The authors should explain what criteria were used to determine what was "the most accurate reflection of training".

Line 164 - what criteria were used to determine a "representative sample of trainers and horses" - potential source of bias?

Line 174 - "assess" - be consistent in tense.

Figure 2 - the singular of "criteria" is "criterion"

Line 296 - sentence starting at the end of this line is ambiguous. “(n=1)" is presumably a reference to the number of papers and not the number of injuries?

Line 326, 368 - the word "incremental" is superfluous

Line 394 - "between-study"

Line 458 - "mission-critical"

Figures - please change the title on each figure so that each is descriptive of the contents.

Round 2

Reviewer 1 Report

The clarifications and removal of the extra information made a rather dramatic improvement to this paper.  

Author Response

Thank you for your review and feedback. We appreciate your comments.

Kind Regards,

Kylie Crawford

Reviewer 2 Report

I am happy with the authors' specific responses to the issues I raised. However, the editorial revisions to this paper appear to have been made in rather a hurry, resulting in the introduction of additional issues. These issues are raised below. 

Lines 140, 148, 197, 202, 214, and elsewhere - “met-analysis” - “meta-analysis” is used elsewhere. Please be consistent.

Line 202 - something is missing in the structure of this paragraph, requiring multiple readings to extract the meaning - please rework.

Figure 4, 5 - labelling has improved but the grouping of tables has made them more difficult to read. Clearly an attempt to harmonise reviewers' comments, but not entirely successful. I assume the journal will provide guidance on this item.

Lines 249 $ 271 - “…suspensory apparatus failure (?)"

Lines 296, 299 - hazard "ratio”, or use HR, but be consistent.

Line 278 - "which" rather than "who".

Line 359 - I didn't mention it last time, but now it stands out more because of the editorial revisions. Purely as a recommendation, I would suggest that the authors not attempt to claim primacy - good science should not be a competition, plus the specifics will always make each paper different in some way.

Line 359 - the authors are once again repeating results in the discussion, which should be avoided. If the grouping of measures now adopted in the paper reveals or fails to reveal some commonality in response for the measures, then this should be addressed directly without repeating results.

Line 380 - almost all of this paragraph is repetition of results or material in the supplementary files. If there is a point to be made, make it without repetition. Basically, the authors appear to be referring to heterogeneity in the data sources used, which would require only one sentence.

Line 391 - this is a little vague. By "discernible" do you mean you did not notice any, or did you perform formal analysis, multivariable analysis, for example? The words "… determine the outcome that could…" should be removed since they only confuse the statement.

Line 398 - influence, not influences

Line 409 - the first part of this paragraph should accommodate the possibility that a second order effect could move in either direction, and that this can only be determined by further analyses within the multivariable analysis that identified the second order effect. This may or may not have been carried out in the studies reviewed. This possibility of differing directions would have a profound impact on interpretation of findings, and thus the interpretation would be relationship- and study-specific. This needs to be made clear rather than have the direction assumed.

Line 457 - comparison, not comparability

Line 467 - it is not clear why the sentence has been struck out - is this a deletion or a move and a legacy of comparison of versions?

Line 472 - confidentiality, not confidentially

The bibliography needs work. There is inconsistency in the citation style.

Reference my first review, point 8, I have now located the offending citation in your bibliography, and this concern is withdrawn.

Author Response

Please see attached point by point response.

Kind Regards,

Kylie Crawford
